# Jesus's Death as Communal Resurrection in Mark Dornford-May's 2006 Film *Son of Man*

Stephen P. Ahearne-Kroll 

Department of Classical and Near Eastern Religions and Cultures, University of Minnesota, Minneapolis, MN 55455, USA; sahearne@umn.edu

**Abstract:** Instead of trying to recreate the ancient life of Jesus, Mark Dornford-May's film *Son of Man* depicts many famous scenes from the gospels, reworked to tell the story of Jesus in the fictitious "Kingdom of Judea, Afrika" with the concerns of local and global poverty, violence, and imperialism. Jesus's life turns when he directly challenges the Judean leadership, and his arrest, torture, and death reinterpret the dynamics of power from first century imperial Rome in brilliantly analogous fashion both for a localized South African setting and for global settings that struggle under violently repressive governments. Jesus's death stands as the focal point of communal resurrection, inspiring Mary to challenge the oppression perpetrated by those in power. Jesus's death serves to express the complexities of international injustice in South Africa and other countries in Africa and around the world, to embolden and unite an oppressed community, and to shine a light on a mother as the leader of this resurrected community.

**Keywords:** *Son of Man*; South Africa; crucifixion; Roman Empire; Jesus films; disappearance; apartheid; global imperialism

## 1. Introduction

Over 125 films about Jesus have been made since the invention of movies.[1] Before 2006, all but one[2] had portrayed Jesus as a white European or North American. Mark Dornford-May[2006]'s (Dornford-May[2006] 2010) film *Son of Man* was shot in South Africa with an entirely black South African cast, and all but a few minutes of the film in Xhosa, one of the official local languages of South Africa, mostly spoken in the Eastern Cape. *Son of Man* is also filmed on location in South Africa (mostly in Khayelitsha, Cape Town, Western Cape, to be precise). Needless to say, the film is an important addition to the canon of Jesus films, if for no other reason than to loosen the stranglehold that whiteness has had on the depiction of Jesus in film and television. But the film is also a deeply thought-provoking interpretation of the traditional Jesus stories that we find in the New Testament. It does not try to represent the canonical gospels but, rather, tries to tell the story of Jesus in a simultaneously local and universal way.[3]

One of *Son of Man*'s more provocative scenes is that of the crucifixion and resurrection, where Jesus's death is thoroughly interpreted in modern terms that are congruent with its setting, while at the same time capturing the same dynamics of Jesus's execution as it likely happened in the first century. This is no small feat since many Jesus movies try to recreate the crucifixion in gruesome terms for the shock value or to map the enormity of Jesus's individual sacrifice onto modern individualistic theological sensibilities reflected in slogans like "Jesus died for me" (personal salvation) or "Jesus died for my sins" (substitutionary atonement), for example. *Son of Man* completely rewrites the events of Jesus's death and resurrection rather than trying to historically recreate them in a first century style. This rewriting of Jesus's death and resurrection changes its meaning by raising up a communal focus on the import of his death and placing the locus of meaning of the resurrection squarely on the public display of Jesus's body on the cross. This alternative vision for the

meaning of Jesus's death is intricately tied to the cultural location of the story, and really could not be depicted with nearly the same impact in Europe or North America. And yet, this alternative vision for the meaning of Jesus's death simultaneously focuses on the local concerns of South Africa and fans out globally to include the many places that struggle with repressive governments and the violence and poverty resulting from repression.[4]

## 2. The Cultural Setting of *Son of Man*: Local or Global?

The film's production location (mostly in the South African township of Khayelitsha), its dominant language of isiXhosa (spoken mainly in southeastern South Africa), and the South African cast and production team initially point to a localized South African telling of the Jesus story. The first scene—the testing of Jesus by Satan—also points to a localized setting. Jesus is depicted in the garb of an adult man going through the Xhosa circumcision ritual called *Ukwaluka*.[5] Later, the film (starting at 22:29)[6] maps Jesus's baptism onto this ritual, thus interpreting his baptism as a coming of age event with no overtones of washing away sins.[7] There are many other places where Xhosa and greater South African culture features prominently in the film, giving the film a distinctly localized inculturation of the Jesus story, especially apartheid-era South Africa. For example, as Jesus's prominence grows in the film through his teaching and miracles, certain key moments are marked with colorful, simply-painted street murals. As Reinhold Zwick points out, "*Graffiti and street art* were popular media of the anti-apartheid movement. They solidified events, structures, and slogans into a striking public image and became an important part of the resistance's identity" (italics original to author; Zwick 2013, p. 113). *Son of Man*'s use of similar iconography clearly evokes this same dynamic. The film does not just mark key moments in Jesus's life; they are integral to the call for the growing movement of Jesus followers to unite and resist the occupying government's violence and oppression, analogous to anti-apartheid resistance efforts in South Africa.

Another example of *Son of Man*'s South African cultural basis can be found in its use of the *toyi-toyi*, which is a dance/chant of resistance "employed by political and labor leaders and domestic workers and mine workers as part of their revolutionary struggle" during both the anti-apartheid and decolonialization efforts by some South Africans (Adejumobi 2013, p. 80). The *toyi-toyi* appears in a very important scene for my argument, namely, the "crucifixion" of Jesus, where Mary leads a group of followers in a *toyi-toyi* in the face of an armed military group, in order to resist the violence that killed her son. I will unpack this scene in more detail below, but the *toyi-toyi* clearly links the film's concerns with those of localized South Africa.

There are many other elements of the film that set it firmly in a South African context, but there are also indications that the film has wider concerns than that of localized South Africa. The film's second scene (starting at 2:48) describes the story world as set in the fictitious "Kingdom of Judea, Afrika", as indicated by the subheading of a British-style television news report in English about social unrest, as the forces of the democratic coalition "invade the settlements" (2:55) in the midst of a multi-year struggle for control between Herode's militia and the insurgents. The purpose of the coalition insurgents' invasion is to bring "peace to the troubled region" (3:08). This scene is the first of many scenes and details within scenes that do not directly map onto South African history. As Gerald West has put it:

> The Channel 7 News report did have familiar sounding elements in it. The no-
> tion of an African country being 'split' by rival factions, with 'internal' control
> being exercised by a local warlord's militia, and an 'external' 'democratic coali-
> tion', 'insurgent' force attempting to enforce 'peace' in 'the troubled region' are
> familiar ways of talking about African conflicts . . . . But not in South Africa.
> (West 2013, p. 5)[8]

Beginning at 7:28, Joseph and a pregnant Mary are shown walking along a beach and then into a town where all residents are being registered at Herode's orders, obviously

interpreting the trip to Bethlehem by Mary and Joseph for the census in Luke 2:1–7. West points out that

> Joseph follows a well-worn path, from the rural areas of the Eastern Province (the traditional home of the amaXhosa) to the shack-settlements of the Western Cape, in search of work in the areas around Cape Town. What is strange, from a South African perspective, is that Mary accompanies him. Under apartheid, wives and children were not permitted to accompany their men to the cities. So this is another indication that we were watching a post-apartheid film. (West 2013, p. 6)

West goes on to link some of the film's depictions of Jesus to South African concerns, but not to the extent where the film truly represents, and is grounded in, an exclusively South African reality. For instance, Jesus's following in the film grows in the context of the shack dwellers, where he calls for unity, collective dialogue, and action to combat the corrupt and oppressive local government. West says, "His invocation of a 'movement' among the shack dwellers resonated with us, for one of the most significant post-apartheid social movements in South Africa today is Abahlali baseMjondolo (the people who live in shacks) . . . and they are particularly well organized in the Western Cape province, where the film is set" (West 2013, p. 15). But West says the film's Jesus falls short because he never offers a "political or economic manifesto" even in a speech that eventually gets him arrested for his political ambitions (beginning at 47:46). So, for West, *Son of Man* clearly evokes localized South African concerns, but falls short of actually engaging with them the way he would hope or in a way that really reflects a South African context. West seems confused and troubled by the movie because of this perceived inconsistent engagement with South Africa,[9] but he is missing the point of these less than clearly South African cultural references. There will be more on this below as I develop my understanding of the context that the film appeals to in narrating its story as it does.

For S. D. Griere, this imprecise mapping is not as troubling and instead "*Son of Man* provides an artistic, cross-cultural interpretation of the Jesus story" for someone outside Xhosa culture (Griere 2013, p. 23). While I agree with Griere on this point, he limits the film's contextual scope by saying that the film, "beautifully and carefully incarnates the story in contemporary South Africa" (Griere 2013, p. 25) instead of recognizing that there is a wider context being appealed to in the film than contemporary South Africa.[10] Jeremy Punt's analysis has a similar expectational subtext of a South African-only cultural setting for the film, as demonstrated by his objection to the fact that there are no whites in the movie: "The absence of white characters in a movie set in a country imprinted by a racial dispensation which established whiteness as the norm through colonization and apartheid does not quite succeed in eliding white agency" (Punt 2013, p. 51). The military dynamics of the governing bodies, the lack of whites, and the almost exclusive use of isiXhosa (an official but not the only official language of South Africa) are three big factors that would indicate some dissonance between the expectation of a South African-only cultural context and some wider African or even global context.

Dornford-May clarifies the context a bit in two comments: "The story is relevant to the whole world today, not just Africa. Christ was born into an invaded society, so it could be set in Iraq just as easily as Africa"; and "Universal stories are just that—universal although their cultural disguises may change. I took a Middle Eastern myth and explained it through what I see around me every day". (Both quotes from De Waal 2008). Although Dornford-May lived in South Africa at the time he made the film, he was born in 1955 in Yorkshire, England and lived in England until 2000, where he was educated and worked in the theater world. He then married South African actress Pauline Malefane (who played Mary in *Son of Man*) in 2002, after which he became a permanent resident of South Africa in 2004. His work in the theater continued in South Africa as he founded the Isango Ensemble as a continuation of the lyric theater company for the Spier Festival (2000). The Ensemble is made up mostly of actors drawn from South African townships around Cape Town. His work in theater that led up to and continued after *Son of Man* shaped his concern for interpreting the Western theater canon based on the cultural dynamics of South Africa.

Dornford-May's personal perspective and history profoundly shapes the *Son of Man*; "what [he] sees around [him] every day" is not just South Africa, but the wider Western world's interaction with South Africa and other African nations. His view is much broader than what West expects of him, and so West's social and cultural location in South Africa is much narrower in scope than that of Dornford-May's.

The film's cultural setting is crucial to understanding its possible meanings, and, for our purposes, the meaning of the death of Jesus as portrayed in the film. Saheed Yinka Adejumobi's analysis of the film from the perspective of empire and utopia ideologies brings the proper complexity necessary to understand the way that Dornford-May plays with cultural dynamics in South Africa and beyond South Africa simultaneously. Adejumobi astutely points out, "Unlike the former days of imperialism, contemporary expressions of empire have no territorial center of power and do not rely on fixed boundaries or barriers. Such a decentering and deterritorializing process ultimately incorporates the entire globe within its realm" (Adejumobi 2013, p. 70). The concerns of South Africa, then, are depicted in the film, but these concerns are not limited to South Africa because the dynamics of imperialism as they were enacted in 2006 up to the present moment are not those of traditional imperialism, which focused on territorial dominance. Modern imperialism transcends spatial limitations and concerns; *Son of Man* seems acutely aware of this dynamic and therefore constructs its Jesus and his gospel around these concerns: "*Son of Man* situates the Gospel outside the introspective language of empire and within new universal narratives of emancipation for the marginalized, breaking down exclusive boundaries of identity and geospatial realities" (Adejumobi 2013, p. 74).

No element of the film demonstrates the simultaneous concern for the local and the global more clearly than the arrest and "disappearance" of Jesus, starting at 1:03:00 in the film. After Jesus's arrest, he is taken away to Caiaphas. Yet instead of a semi-public trial before the Sanhedrin as depicted in the gospels, Jesus is taken down a long flight of winding steps to a small room where he is interrogated, beaten, and killed by Caiaphas's ruling thugs. After his death, his body is thrown in the trunk of a car, taken to a remote location, and buried in an anonymous grave. Although the extent to which Jesus was tortured, killed, and buried was not always the experience of those who ran up against the apartheid regime in South Africa, the pattern of arrest without charge, detention for undefined terms, and disappearance was common for activists during apartheid era South Africa.[11] The regime's actions regarding its treatment of anti-apartheid activists became an important focus of the Truth and Reconciliation Commission's proceedings in the post-apartheid era.[12] The Commission's testimonies were often the only way to learn about the brutality of the South African police and authorities because as soon as it was clear that a new, Nelson Mandela-led government was quickly arriving, "wholesale burning of files began" (Goodwin and Schiff 1995, p. 565).

The most famous "disappearance" of the apartheid era was that of Steve Biko, the charismatic anti-apartheid leader known for the creation and leadership of the Black Consciousness movement as a way of advocating for equality between whites and non-whites in South Africa. Purportedly, Biko said to Thenjiwe Mtintso in 1977, "These guys—the day they get me—they'll kill me, because I'll beat up the guy or make him beat me so that I just die. If my hands are tied, I will spit in his face. I'm not going to answer questions that I don't want to answer" (Goodwin and Schiff 1995, p. 565). Biko died shortly after he made this statement. Here are the basics of his last days: On 18 August 1977, Biko and his friend Peter Jones were stopped at a roadblock in the Eastern Cape and arrested for violation of the order that banned him from traveling outside King William's Town. Jones and Biko were separated, with Biko being transferred to the Walmer police station in Port Elizabeth and then to another facility in central Port Elizabeth on 6 September. There he was interrogated for 22 h, while being handcuffed, shackled, and chained to a grille (Bucher 2012, p. 569; see also Mangcu 2014, p. 260). While the details are unclear as to what precisely happened while Biko was in custody during this interrogation, he seems to have been severely beaten and suffered massive head trauma. While several doctors examined

him without recommending further care, the last two agreed he should be transferred to a prison hospital in Pretoria (Bucher 2012, p. 569), some 1190 km away. Police then put him, naked, in the back of a Land Rover on 11 September and drove him to the prison hospital where he died in his cell the next day on 12 September (Mangcu 2014, pp. 261–62). Despite many efforts to cover up the wrongdoings of the interrogators by police, official doctors, and political officials (the official line was that his death was caused by a hunger strike), an autopsy revealed the extent of the blunt force injuries to his brain, indicating death by beating. Eventually the five police officers who were on duty the night Biko was killed were investigated by the Truth and Reconciliation Commission from 1996–1998. Although their stories conflicted with one another, they were all implicated in Biko's death. However, they were never prosecuted because the statute of limitations had expired. The similarity between Biko's death and Jesus's death in *Son of Man* is clear, as has been pointed out by many commentators (West 2013, pp. 17–18; Adejumobi 2013, p. 84; Zwick 2013, p. 113; Walsh 2013, p. 198; Mokoena 2017), and firmly grounds Jesus's death in the concerns of apartheid-era repression of resistance activists.

Yet the phenomenon of "disappearance" is not limited to South Africa, even disappearances that end in torture and death, as Biko's did. Globally, the use of disappearances by repressive or autocratic governments is widespread. On 18 December 1992, the United Nations General Assembly adopted resolution 47/133, which says that enforced disappearance occurs when:

> persons are arrested, detained or abducted against their will or otherwise deprived of their liberty by officials of different branches or levels of Government, or by organized groups or private individuals acting on behalf of, or with the support, direct or indirect, consent or acquiescence of the Government, followed by a refusal to disclose the fate or whereabouts of the persons concerned or a refusal to acknowledge the deprivation of their liberty, which places such persons outside the protection of the law. [13]

Although enforced disappearances certainly occurred in South Africa during the apartheid era, other African nations also have been known to wield it, such as Liberia, Sierra Leone, Algeria, Rwanda, Ghana, and Sudan (Griere 2013, p. 25n.6). Yet the phenomenon is not limited to African nations. Again, according to the United Nations,

> Enforced disappearance has become a global problem and is not restricted to a specific region of the world. Once largely the product of military dictatorships, enforced disappearances can nowadays be perpetrated in complex situations of internal conflict, especially as a means of political repression of opponents . . . .Hundreds of thousands of people have vanished during conflict or periods of repression in at least 85 countries around the world. [14]

On 30 August 2021, to mark the International Day of the Victims of Enforced Disappearances,[15] the UN Secretary-General António Guterres said that "enforced disappearance continues to be used across the world as **a method of repression, terror, and stifling dissent**. '**Paradoxically, it is sometimes used under the pretext of countering crime or terrorism.** Lawyers, witnesses, political opposition, and human rights defenders are particularly at risk'".[16] As of the release of *Son of Man*, there were already two UN resolutions regarding enforced disappearances, 33/173 (20 December 1978) and 47/133 (12 February 1993), with one more to come shortly after the release of the film (61/177 adopted on 12 January 2007).

Accordingly, the film simultaneously evokes the local context of disappearances in apartheid-era South Africa *and* the phenomenon of disappearances in military dictatorships, repressive governments, and "complex situations of internal strife" around the world, as the UN describes in the quote above.

### 3. Jesus's Death in *Son of Man*

I would like to pick up on the first part of the boldfaced statement from the UN website: "**a method of repression, terror, and stifling dissent**". Enforced disappearance is really a form of government sponsored and executed terror. Rev. G. De Klerk's testimony to the Truth and Reconciliation Commission on 19 February 1997 captures the dynamic well: "The regime could not arrest all the protesting people and focused themselves on eliminating the leaders".[17] By targeting resistance leaders and activist leaders, repressive governments can seek to control resistance by terrorizing its members. The explicit message is that if resistance continues, the same thing that happened to the resistance leaders can happen to any of its members. Terror is an efficient means of control because it uses a relatively small amount of resources to control a large group of people. In the hands of a government, it can be wielded most effectively to keep control over its people and maintain power for the governing authorities. Although Giere's characterization of this sort of use of "disappearance" as a "governing strategy" (Griere 2013, p. 25) in repressive countries is accurate, he does not capture the terror that it inflicts on the populations to whom it is directed. Enforced disappearance is not just a governing strategy but a way of maintaining power by crushing the will of the people by, in turn, crushing the bodies of the leaders who disappear.

As I have already described, Jesus's death in *Son of Man* comes in the form of an enforced disappearance and not by crucifixion. As far as I know, up to 2006 *Son of Man* is the only Jesus film that depicts the death of Jesus in any other way than crucifixion.[18] And yet, the death of Jesus in the film is entirely appropriate for its setting and highly effective for the film's rendering of the Jesus story. Why? In order to answer this question, we need to explore Jesus's death as portrayed in the canonical gospels to get a sense of how Dornford-May might be interpreting the gospels with respect to Jesus's death. Griere says, "Disappearance is the primary interpretive lens for the crucifixion" (Griere 2013, pp. 28–29). Yet the reverse makes just as much sense: The crucifixion is the primary lens for disappearance.

Roman crucifixion was a brutal way to be executed, although descriptions of crucifixion are not widely available, especially the particulars of how a person was attached to scaffolding that constituted the act. To give a sense of the brutal possibilities, let me turn to a mime performed for the first time in 41 C.E., as part of the *ludi Palatini*, which was a private festival held in honor of the deified Augustus. The mime told the "exploits of the notorious Roman bandit Laureolus" (Harley 2019, p. 303). In the mime's first performance, a copious amount of fake blood was used to depict the crucifixion of Laureolus to dramatize the brutality. Martial refers to a performance of the mime a few decades later where an actual criminal was procured, and a live execution was performed on stage. The criminal was suspended from a real cross where he was then mauled by a bear. Martial says of the criminal, "His mangled limbs were still alive, though the parts were dripping with blood, and in his whole body there was actually no body".[19]

Historians who have studied Roman crucifixion—or more generically "how Romans suspended people" as Harley puts it (Harley 2019, p. 307; see also Chapman 2010, p. 32, from whom Harley draws)—have grappled with the difficulty of defining it precisely. Seneca not only describes the large numbers of crucifixions during uprisings, but also the variety of techniques for it, witnessing, in turn, the variety of ways to understand *crux*.[20] Other literary evidence varies and oftentimes is contradictory. The archaeological and iconographical evidence only adds minimal information to our understanding of the practice. Curiously, "As the single most comprehensively documented extant record of an execution by this method, [the canonical Gospels] constitute our most helpful sources of evidence for the carrying out of a crucifixion in Roman antiquity" (Harley 2019, p. 308). Even the canonical Gospels, though, are not exactly detailed in their narrations of Jesus's death, with respect to the mechanism of crucifixion. Only the Gospels of John and Luke clearly state that Jesus's hands and feet were pierced in the process.[21] Yet using the Gospels has its own difficulties and so they must be used as case studies and not as the exempla of crucifixion in the Roman world.[22]

Cook has performed a balanced and thorough treatment of the evidence in his body of work during the teens of this century. He comes to these measured conclusions about the practice of crucifixion (as summarized by Harley 2019, p. 317):

> [T]he penalty was used against a variety of individuals, including slaves and criminals; crucifixions were carried out by a public executioner or a military authority such as a centurion; various forms of torture could precede it; victims were often walked in chains to their place of execution, often designated outside the city; they might be "patibulated", carrying the horizontal part of their cross to that place, where the vertical beam would already be in place; victims could be stripped, but were not necessarily naked; ropes or nails or perhaps both could be used to affix them to their instrument of torture; they could be upright or in different poses, such as upside down; the magistrate read the charge from a placard or titulus and this could then be placed on the cross; bodies could rot on crosses or be buried. (Cook 2014, pp. 423–30)

This is about as specific as one can get about the particulars of crucifixion, but it is also clear that it is one among many techniques of suspension as a form of torture and execution as a punitive act in the Roman Empire. It was an act of execution "in the culture of public spectacle" (Harley 2019, p. 322).

This culture of public spectacle was not merely for the masses of Romans wanting to be entertained, however masochistically one might imagine this entertainment. It was part of a greater culture of torture and terror directed toward the lowest class in Rome and those in the provinces who crossed the Empire by challenging its authority. Along with Seneca *Dial* 6.20.3 (mentioned above), Josephus speaks of the use of crucifixion frequently in *Jewish Wars* (2.75; 2.241; 2.253; 2.306, 308; 3.321; 5.289; 5.449–451) and *Antiquities* (17.295; 20.129) to suppress rebellion in the provinces (see also Hengel 1977, pp. 46–47). In *Against Verres*, Cicero talks about the use of crucifixion as a punishment for high treason by a Roman citizen (2.5.158–165), but this was not the norm. Crucifixion was more commonly used against slaves, sometimes just for the sport of it. Although it is notoriously difficult to access the lives of slaves in ancient Rome, L. L. Welborn has argued that depictions of slaves in popular comedy, satires, and novels can help communicate "[h]ow deeply slaves lived in the shadow of the cross" (Welborn 2013, p. 135). In contrast, in the literature of the elite, there is almost no reference to crucifixion. Turning to Cicero again, this time in *Pro Rabirio*, he says that "the very word of the cross should be far removed, not only from the person of a Roman citizen, but from his thoughts, his eyes and his ears . . . . The mere mention of such a thing is shameful to a Roman citizen and a free man" (5.16; see Welborn 2013, p. 135 for the quotation and translation). This attitude seems to have been internalized by the elite, but even when crucifixion is mentioned by Greek and Roman historians, for example, "there is a reticence and tendency to portray crucifixion as a barbarian mode of execution" (Welborn 2013, pp. 135–36; see also Hengel 1977, p. 23). Despite this elite attitude towards crucifixion, it was ubiquitous in the culture of public spectacle in Rome, across Italy, and the Empire as a punishment for slaves. It is worth quoting Welborn's long synopsis to get a sense of the horror of crucifixion for slaves:

> Just outside the Esquiline Gate at Rome, on the road to Tibur, was a horrific place where crosses were routinely set up for the punishment of slaves. There a torture and execution service was operated by a group of funeral contractors, who were open to business from private citizens and public authorities alike. There slaves were flogged and crucified at a charge to their masters of 4 sesterces per person . . . . Varro mentions rotting corpses; Horace speaks of whitened bones; Juvenal describes the way in which the Esquiline vulture disposed of the bodies . . . . An inscription from Puteoli confirms that such places of execution, with crosses and other instruments of torture, were found throughout Italy and probably outside the gates of every large city in the Roman Empire. At these places of execution, it is impossible not to recognize the real reason for the silence of the upper class with respect to crucifixion: crucifixion was the "slaves' punishment"

(*servile supplicium*). (Welborn 2013, p. 136, with references to the primary sources in n.95–102)

Crucifixion, then, especially for the enslaved class in the Empire, was a constant threat, a terrorization of those with the least power in society, and a major means to control the population from which the power of the elite, ruling class was constructed.

The association of crucifixion with punishment, terror, and death of the enslaved affected other classes, as well. Consider the repulsion that Cicero had for the practice in the quote from *Pro Rabirio* above. In his discussion of the treasonous citizen in *Against Verres* 2.5.158–165, Cicero narrates that the man in question repeatedly cried out that he was a Roman citizen as he was beaten, tortured, and crucified. Cicero laments the offence—not the crime of the man, but the affront on his Roman sensibilities that a citizen would be treated as such:

> Have our rights fallen so far, that in a province of the Roman people—in a town of our confederate allies—a Roman citizen should be bound in the forum, and beaten with rods by a man who only had the fasces and the axes through the kindness of the Roman people? . . . If the bitter entreaties and the miserable cries of that man had no power to restrain you, were you not moved even by the weeping and loud groans of the Roman citizens who were present at the time? Did you dare to drag any one to the cross who said that he was a Roman citizen? (2.5.163)[23]

As Cicero describes the horror that he felt at the site of a citizen being tortured and crucified, we can imagine the horror that slaves and non-citizens felt at the seemingly unfettered power that Roman authorities wielded to use this instrument of terror against them. The threat of crucifixion seemed to hold all in its grips, but none more than the masses that needed to be controlled in order for Rome to maintain its power.

Jesus—a lower-class, non-citizen—suffered under this threat and succumbed to the power of Rome with his death by crucifixion. The death of the Jesus in *Son of Man* matches the death of the earthly Jesus, not in detail but in effect because the mode of terror and control that crucifixion enacted for the Romans matches the mode of terror and control that disappearances enacted for the authorities in *Son of Man*. Dornford-May's choice of this mode of execution for Jesus not only perfectly captures the social pathos of the earthly Jesus's execution,[24] but it also expresses this reality in global terms, given the widespread use of disappearance across the world by repressive governments and during times of strife and war. As West puts it, when Mary finds out that Jesus was buried and not in prison somewhere, "Jesus becomes the son of all the many thousands of South African women who to this day do not know where their sons, detained and killed by the apartheid regime, are buried" (West 2013, p. 18; see also Griere 2013, p. 25). Adejumobi widens the scope a bit in his assessment of Jesus's death: "While the visuals of the assassination and secret burial of Jesus evoke the demise of the Congo's first postcolonial prime minister Patrice Lumumba, carting Jesus in the truck of a car also reminds the postcolonial scholar of Steven Biko, the activist and pioneer of 'Black Consciousness' youth resistance, who died in the custody of apartheid government. *Son of Man* remembers not only these leaders' untimely demises but also their legacy in the African quest for freedom" (Adejumobi 2013, p. 84). Indeed, there is significant overlap between the depiction of Jesus's demise in *Son of Man* and the arrest and death of Biko, but I would extend this legacy to all the leaders (and others) who have lost their lives in the over 85 countries that the U.N. identifies as perpetrating similar disappearances. Dornford-May's quest to tell the universal story of Jesus through a localized setting is instantiated in the way that Jesus's death in *Son of Man* speaks to the global realities of countless individuals, families, and communities who have suffered through disappearances. The significance of Jesus's death in the localized context of Roman Judea in the larger, "global" Roman Empire was also universalized quite early in the development of Christianity, so Dornford-May's interpretation of his death for his own film's setting expresses the same universal significance, but for a modern context.

#### 4. The Displacement and Redefinition of the Crucifixion in *Son of Man*

Dornford-May does not completely reject the crucifixion in his film, however. Jesus is crucified, but not as a means of death. After Mary finds out where Jesus is buried (starting at 1:11:05), she is taken to the place of his burial and mourns his death with several women companions and Hundred, the film's centurion. Hundred witnesses Jesus's burial but has a change of heart and tells Mary where the body is. As the others are getting ready to leave the burial site, Mary begins to dig up the body of Jesus with only her hands. She eventually succeeds and carries the body of Jesus back to town in the back of a truck, filmed in a 12-s shot that evokes Michaelangelo's *Pietà* (starting at 1:12:37). That night, Mary ties Jesus's body to a cross and displays it on a guard tower overlooking the town. At daybreak, she stands before the cross, then she turns to face the residents, as they notice her and begin to silently and solemnly come towards the tower. As Mary looks at the crowd, she begins to sing, "This land is covered in darkness", in chorus with her companions. The crowd joins in, and some, including Jesus's disciples, join Mary on the platform. The song develops into a dance and the participants call "Comrades unite . . . Unite, freedom fighters! Strength, comrades!" as a helicopter circles overhead and troops show up soon after to disperse the crowd. The scene has intermittent shots of the body of Jesus displayed on the cross in the background of the powerful crowd chanting and dancing on the platform in a *toyi-toyi* dance, which evokes "the anti-apartheid and decolonization period . . . . [which] has been employed by political and labor leaders and domestic workers and mine workers as part of their revolutionary struggle" (Adejumobi 2013, p. 80).

When the troops arrive (1:17:30), its leader—a youthful figure in a floppy hat—orders the crowd to disperse. His orders have no effect. Mary continues to lead the *toyi-toyi* with her back to the troops, occasionally chanting over her shoulder toward them. The leader gives the order to fire, and one of the soldiers fires over the heads of the group on the platform, which scatters the crowd around the platform and causes those on the platform to fall to their knees. After a few seconds, Mary rises, turns around to face the soldiers, looks at the body of Jesus, back at the soldiers, walks closer to the soldiers, and begins the *toyi-toyi* again, this time facing them, with all those on the platform now on their feet chanting and dancing with her, with Jesus on the cross in the upper right hand corner of the screen and the landscape of the shack settlements stretching out in the background. As she begins the *toyi-toyi* again, the shot includes the guns of two soldiers in the foreground, but as the scene continues, the camera slightly zooms in on the group and the guns disappear. Mary and her female companions lead the crowd. The scene is then immortalized on a community mural in the next shot (1:20:03).

I describe this sequence in some detail because of its power and because of the confluence of the film's themes present in it. Many have commented on how the "crucifixion" of Jesus functions to unmask the evil of disappearances by revealing the truth of the atrocity, undoing the effectiveness of disappearance by making the victim present again, and thus, undoing the terror that was inflicted by the governing body (Zwick 2013, pp. 113–14; West 2013, p. 18;[25] Griere 2013, pp. 25, 31; Nadar 2013, pp. 62–63; Adejumobi 2013, p. 83;[26] Roher-Walsh and Walsh 2013, p. 173; Baugh 2013). I agree with this assessment. Because the film depicts the crucifixion this way, the public spectacle of crucifixion at play in ancient Rome is separated from the terror and control that crucifixion enacted for the Romans. Indeed, the public display of the crucifixion was integral to the Romans' efforts to instill terror and control. This difference signals a major shift in Dornford-May's depiction of Jesus's death in comparison with ancient crucifixions. With the choice to displace the crucifixion until after Jesus's death as an act of protest and uncovering of injustice, Dornford-May effectively transposes onto the cross the traditional ways of interpreting the resurrection in the New Testament. The resurrection is often seen by scholars and Christians as the vindication of Jesus and God, proving that his conviction and death for the crime of being King of the Jews was unjust, and so, Jesus was wrongfully executed.[27] Certainly, this way of understanding the resurrection is valid, but it focuses too much on the individual of

Jesus, an individualism that *Son of Man* resists throughout the film and one that did not exist in the first century Mediterranean.

Jesus's message in the film was never about himself but about the kind of society he envisioned and the challenge he posed to his disciples to live out that society. At 29:23, after Jesus gathers his disciples, in the first scene of their meeting, he talks of not standing on the sidelines and offers a social analysis of the causes of unrest that the interim government uses as a pretense for its presence. "Unrest is due to poverty, overcrowding and lack of education", Jesus says. But he goes on to say, "We must prove to them that we are committed to non-violent change . . . Each human life is important. It is our right to protect our beliefs . . . but this never becomes the right to kill". Jesus begins the transformation of society that he envisions by demanding that his group rid themselves of their weapons even in the face of ongoing governmental violence, including the weapons of Judas, who is shown with a flashback to his days as a kidnapped child soldier-in-training, asked to kill his first prisoner as a member of the force (30:55). Clearly, violence is deeply ingrained in the culture of Jesus's first followers. Then he welcomes, heals, and raises those in most need, while also teaching about the evils of global economic imperialism (40:40). All the while, from the beginning of the movie until the speech that gets Jesus arrested (48:10), working together, comradeship, and communal unity are a constant refrain from Jesus and the angels that support his work. In that final speech, Jesus says, "Let us work together, because through collective dialogue, we can penetrate the deafest of ears. It feels like we are defeated. We need to act as a movement to ensure each of us is treated with dignity. Let us unite. Solidarity! Unity!" (48:10–48:44). Jesus saw himself as a representative and unifying force to help the people collectively overcome injustice and transform the community and its leadership.

Jesus's post-mortem crucifixion continues his work. His death initially leads to the mourning and dissolution of the movement, but when Mary and her companions find Jesus's body, raise it from the grave, and display it on the cross, the community reassembles around the cross at the guard tower. That reassembly leads to reunification, initially to mourn and pay homage, but as Mary begins singing softly with her eyes closed, almost to herself, "This land is covered in darkness", others join in, and Mary opens her eyes, singing to and with those gathered. Dornford-May shoots this frame with a frontal close-up of Mary and a blurred Jesus behind her left shoulder, his head hanging limp. Mary looks alive, defiant, and strong as her voice grows louder. Others join her on the platform and the singing grows. She is the clear leader of the Jesus movement (reinforced by repeated shots of her singing with Jesus in the background). The singing becomes a *toyi-toyi* as they chant rhythmically, almost with a focused ecstasy, "Comrades unite!" and "Unite, freedom fighters! Strength, comrades!" When the soldiers arrive, the crowd disperses, all except those on the platform. They continue to sing and dance as one. After the soldiers fire overhead and silence the dance, Mary looks at her dead, 'undissappeared' son and starts the *toyi-toyi* up again facing the soldiers, with the implication of a challenging invitation to join, just like she faced the crowd and inspired them to unite.

The community is resurrected and empowered to continue Jesus's work to unite and transform their society through the leadership of Mary and her companions. The community is not dead because its leader is not dead. He is resurrected in Mary and her companions.

## 5. Conclusions

The final scene of the film's fictitious setting[28] is an appearance of Jesus walking up the hill towards the guard tower surrounded by the child angels and hand-in-hand with Gabriel as joyful music rings out in the background. The cross is draped with the red ties that Mary used to secure Jesus's corpse to the cross, but otherwise the cross and the tower platform are empty. Mary and her companions are no longer there. When Jesus reaches the base of the tower, he turns around, faces the angels, and thrusts his hand up with a triumphant shout and a huge smile. While some see this as the individual resurrection of Jesus, I see

it as signifying the joy that Jesus and the angels have because of the resurrection of the community and the continuation of the unification and communal transformation that he pushed for when he was alive. Mary and her companions are no longer on the platform but are off together, unified in their fight for justice and dignity. Mary and the disciples understood what Jesus was trying to do and teach, and their unification to continue the struggle together is the triumph of the resurrection. For Dornford-May, this struggle is both local to South Africa and universal to all those places that struggle under the repressive regimes that rob their populations of dignity and life.

**Funding:** This research received no external funding.

**Institutional Review Board Statement:** Not applicable.

**Informed Consent Statement:** Not applicable.

**Conflicts of Interest:** The author declares no conflict of interest.

## Notes

1. I wish to express my thanks to Micah Meyer for her assistance in researching the phenomenon of disappearance for this article, and to the two anonymous reviewers who gave many helpful suggestions that made this a stronger article.

2. *Ocean of Mercy* (*Karunamayudu*) (dir. Bhim Singh 1978) is an Indian film shot entirely with an Indian cast and crew in India. Jean-Claude La Marre's *The Color of the Cross* was released in 2006 (as was *Son of Man*) and depicted Jesus as a black man whose death was a racially motivated hate crime. The Italian film *Black Jesus* (*Seduto all sua destra*, Valero Zurlini 1968) is not a Jesus movie per se. It tells the story of a fictious rebel leader who is arrested by the military, tortured, and killed, which turns him into a martyr for the movement. The story is inspired by the arrest and death of Patrice Lumumba, Congo's first democratic leader.

3. If the reader has not seen *Son of Man*, I highly recommend viewing the film before continuing with this article. Not only will the article make more sense, but the film is well worth watching.

4. The film is bookended by two scenes that are set in the cosmic realm, possibly couching the whole film a cosmic struggle between good and evil. I think this is too simplistic of a reading of the film, however. The film opens with the testing of Jesus by Satan, with Jesus in the garb of the Xhosa circumcision ritual called *Ukwaluka.* The last scene before the credits (which have their own series of images) is that of Jesus and the child angels after his resurrection. These two scenes couch the film in the cosmic realm as a confrontation between heavenly and Satanic forces. Peppered throughout the movie are appearances of the Satan figure (holding the deer hoof cane) to remind the audience of the ongoing confrontation. However, every reference to the cosmic realm has an anchor in the local, earthly realm. The opening scene has Jesus and Satan on an earthly sand dune with Jesus dressed for and ready to enter into the *Ukwaluka*. His final rebuff of Satan is "This is my world!" and he is not talking about the cosmic realm. This is reinforced when the boy Jesus reiterates this conviction after the slaughter of the innocents, when the angel Gabriel offers to take Jesus away from the horror of the slaughter (22:00). And the final scene of the film has the resurrected Jesus with the angels, but they all appear climbing the hill leading to the guard tower where his now empty cross remains. All of these scenes reinforce a local, earthly, contingent emphasis of the film, rather than the cosmic, even while interweaving the cosmic with the earthly throughout the film (e.g., the repeated presence of the Satan figure, but also the companionship of the angels for Jesus, visible only to him and the audience). See (Baugh 2013, p. 127) who briefly treats the cosmic struggle depicted in the film.

5. See (Griere 2013, pp. 25–27) for a discussion of the ritual.

6. All time references to *Son of Man* correspond with the streamed version of the film.

7. Jesus's baptism as depicted in Matthew, Mark, and Luke can certainly be seen as a transition to Jesus becoming the adult Son of God, but all three gospels depict Jesus's baptism as performed by John, whose baptism is one of repentance for the forgiveness of sins. Jesus is not baptized in the Gospel of John.

8. West's essay, as a whole, is an instructive reading of the movie from a South African biblical scholar's perspective, but he spends his time looking in vain for qualities and messages of *Son of Man* that speak directly and precisely to his situation and experience of South Africa. In the end, he appreciates some parts of the film but seems disappointed and frustrated that the film did not present the Jesus he wanted and a Jesus who spoke more directly to South African politics and culture. His essay misses the point of the movie, in my opinion.

9. "So much of the film is suggestive, but it is frustratingly difficult to locate the film within our current context. Is the son of man really 'a South African' son of man? The echoes are everywhere in this film, but finding a coherent biblical or contextual trajectory is difficult. Perhaps all it is offering is 'prophetic fragments'" (West 2013, p. 16).

10. I will develop this more below, but at the very least there are the following contexts that arguably come into play in the film: contemporary (2006) South Africa; apartheid South Africa; other regions of Africa that struggle with authoritarian government systems; and global contexts that suffer under violently repressive governments.

[11]　A July 2019 article in U.S. News & World Report, which talks about a reopened case regarding the death of activist Ahmed Timol in October 1971, says, "The precedent-setting decision [about Timol's case] may now open the door to more investigations into the deaths and disappearances of dozens, possibly hundreds, of other activists" (see "Inquest Revives the Pain of Apartheid-Era Deaths", *U.S. News & World Report*, 4 July 2019).

[12]　The Truth and Reconciliation Commission (TRC) began in 1995; its main purpose was "to promote reconciliation and forgiveness among perpetrators and victims of apartheid by the full disclosure of truth" (see https://www.apartheidmuseum.org/exhibitions/the-truth-and-reconciliation-commission-trc; retrieved on 6 June 2022). See also the *Sunday Times* "Heritage Project": "The TRC was mandated with the investigation of gross violations of human rights, torture or extreme ill treatment, murder or its attempt and kidnapping or 'disappearance' between 1 March 1960 and 11 May 1994" (https://sthp.saha.org.za/memorial/articles/the_truth_and_reconciliation_commission.htm; accessed on 6 June 2022).

[13]　See https://www.un.org/en/observances/victims-enforced-disappearance; retrieved on 6 June 2022.

[14]　https://www.un.org/en/observances/victims-enforced-disappearance; retrieved on 6 June 2022.

[15]　This day of remembrance was established by the UN General Assembly's resolution 65/209 on 21 December 2010 to be celebrated every 30 August beginning in 2011.

[16]　https://news.un.org/en/story/2021/08/1098702; retrieved on 6 June 2022. Bold type is original to the website.

[17]　Testimony to the Truth and Reconciliation Commission by Rev G. De Klerk (19 February 1997) https://www.justice.gov.za/trc/reparations/oudtshoo.htm. Accessed 2 June 2022.

[18]　*Jésus de Montréal* (1989) has another form of death for the Jesus-like character, but the film itself is not a rendition of the story of Jesus that follows the typical plot line for Jesus's life as told in the gospels. It is more akin to *The Matrix* (1999), which has many allusions to Jesus and the events of his life, death, and resurrection but is not a retelling of the Jesus story. *Godspell* (1973) depicts Jesus's death not as a literal crucifixion, although he dies with arms outstretched tied to a chain-link fence, certainly evoking the crucifixion.

[19]　*uiuebant laceri membris stillantibus artus | inque omni nusquam corpore corpus erat.* Translation (Harley 2019, p. 304).

[20]　See Seneca *Dial.* 6.20.3 and Cook's (2014, pp. 34–35) discussion of it.

[21]　In the second part of his resurrection appearance, Luke's Jesus shows the disciples his hands and feet to prove that it is he and that he is alive (see Luke 24:38–40). John insinuates the method of crucifixion in the upper room, when Jesus shows Thomas his hands and side as proof that he was both alive and the one who was crucified (see John 20:26–28).

[22]　As Harley points out, "This immediately becomes problematic: the experiences of Jesus are routinely upheld or presented as the 'norm' or model, rather than as a case study, of Roman practice, with the gospel accounts interrogated as descriptions of a historical event. Given that those accounts also constitute the earliest theological interpretations of that event, it is necessary to proceed with some caution" (Harley 2019, p. 309).

[23]　Translation by C.D. Yonge 1903. Accessed on 8 June 2022 at https://www.perseus.tufts.edu/hopper/text?doc=Perseus%3Atext%3A1999.02.0018%3Atext%3DVer.%3Aactio%3D2%3Abook%3D5%3Asection%3D163.

[24]　Against Griere, who states, "[T]his may be the point where the telling of the Jesus story in *Son of Man* builds the most friction with the New Testament gospels and Christian doctrinal orthodoxy" (Griere 2013, p. 25). While Griere is likely talking about the details of the Jesus stories in the gospels, I am referring to the necessary interpretation that must go into making a Jesus film. Most Jesus filmmakers are not so open about their interpretive choices, but Dornford-May is, and it is spot-on for the global context he is envisioning, in my opinion.

[25]　"We are left with Mary and the other women of South Africa, who, like Rizpah, care for the dead and in so doing shame the forces of patriarchy, political dictatorship, and military power".

[26]　"It is also the resurrection of the agency of citizens who are capable of writing themselves into a universal (p. 84) epic of global justice".

[27]　I actually think that the way that the gospels narrate and interpret the resurrection is quite varied and nuanced. Mark, for instance, does not have a triumphal resurrection but an empty tomb. Matthew's resurrection is one that continues Jesus's presence for the spreading of the gospel. John's resurrection is the final "sign" that Jesus is from above. Luke comes closest to the resurrection being a vindication of Jesus, and we see that in the strong overtones of his death being that of an innocent man (see especially Luke 23:47, "Surely, this man was innocent", declares the centurion). All the gospels emphasize that even though Jesus was raised, he remains in his resurrection the one who has been crucified.

[28]　As the credits roll, the film shows scenes of actual children and families in a South African township. But this scene is no longer part of the fictitious story world of the film.

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
