# Peer review of "Jesus’s Death as Communal Resurrection in Mark Dornford-May’s 2006 Film Son of Man"

_religions, doi:10.3390/rel13070635_

Round 1

Reviewer 1 Report

Overall this is an interesting and insightful analysis that needs just a little editing and expansion to be ready for publication.  The overall approach and writing are fine. 

Serious factual error at line 195. The author notes, "Biko died shortly after on August 18, 1977." That is wrong.  Biko was arrested in the early morning hours of 18 August, spent almost a month in police custody, and died on 12 September 1977.  The rest of the information regarding the autopsy and the later findings of the TRC are correct, but that is a serious error of fact.

Given the length of time the author spends on Biko, more must be done to connect Biko as Christ-like figure.  While I understand what the author is doing and saying, I think she/he needs to be more explicit in the Biko/Christ connection - the idea of the suffering martyr for his people as well as the obvious reference to Biko in Son of Man's execution scene.  There is more to be found in that analysis.

Note 16 - Yay!  I appreciate the mentions of both Godspell and Jesus of Montreal (a personal favorite), but humbly suggest the latter be referred to by its actual, original title, Jésus de Montréal - it is after all, a Quebecois film and we should be respectful of that culture as well. 

If, per line 320, crucifixion is part of a "culture of public spectacle," then what are we to do with the very private execution of Jesus in this film?  I would urge the author to think about this contradiction and perhaps find the public spectacle aspect is found in the re-crucifixion of the body in public, so to speak.  Since the death was done in private, it is "re-enacted" in public, something the author hints at but does not fully explore.  Similarly, around lines 345-375, the exploration of crucifixion as "slave execution" has more to explore, considering finding a more direct link to enforced disappearance as the contemporary version of "slave execution," the key difference being that the one is public and the other private - crucifixion is about public displays of power and enforced disappearance is about removing the body of the person from the public sphere permanently.  There may be some valuable ideas found in Foucault's Discipline and Punish in terms of how the state treats the bodies of "undesirables."  The implication being, of course, that modern regimes don't crucify as it is "bad pr" for a modern nation to torture bodies, so instead the absent body becomes the sign of state power.  Mary, at the end of the film, defies that power by bringing the actual body back.

Otherwise a great and interesting essay that will make a worthwhile addition to the special issue.  I look forward to seeing a revised version. 

Reviewer 2 Report

Reviewer comments: Jesus’s Death as Communal Resurrection in Mark Dornford-May’s 2006 film Son of Man

General Comments:

·       This essay uncovers a range of cinematic/aesthetic strategies through which Mark Dornford-May’s Son of Man authentically transposes the passion onto a contemporary reality in both local and universal ways. Its reading of the film is well informed and demonstrates a clear and lucid understanding of the South African cultural and political context. Its most interesting, and perhaps most important contribution lies in its original and important explanation of the relationship between crucifixion and disappearance as a mode of terror and control as well as its sustained reading of the final scenes of the film and their political/social subtexts and contexts.

·       The overall argument would benefit from some refinement and a further contextualizing of the film within the Jesus film tradition’s broader treatment of the passion. While Dornford-May is perhaps the most radical updating of the passion to a contemporary context, there are resonances in the cinematic tradition, which I highlight in the specific comments and perhaps the essay could be strengthened by incorporating some of these instances.

·       Some of the analysis, for example, points to a universal/local dynamic at play in the film. Some greater attention could be paid to the ways in which Dornford-May emphasizes the transcendent as a bookending dimension of the film. These are implied but could be emphasized in a more sustained manner. In my specific comments, I have suggested some instances in the text where this might be appropriate.

·       The writing style could be refined further and I have highlighted some stylistic issues in the comments below. While these are largely minor, they would improve the readability of the essay. On occasion, there is the odd word missing or word repetition. Some word choices (such as ‘waffling’) should be revised.

·       Most of these comments are at the level of suggestions and, in my view, would represent moderate alterations to the text. In light of this, I would suggest a minor revision of the text before publication.

Specific Comments:

Page 1

It is worth noting that The Color of the Cross, directed by Jean-Claude La Marre, was released in the same year and focused on Jesus' death being, partly at least, the result of racism.

Word choice: in par. 1 the film is described as a 'welcomed addition' - should this be a 'welcomed addition'?

Sentence structure: in par. 1 at the end, it might be better to formulate the sentence as: 'It does not try to represent the canonical gospels but, rather, to tell the story of Jesus in a [note word missing] simultaneously local and universal way.'

Missing words: in par. 2, line 32, it might be better to say: 'that are congruent with . . .'

Broader reference: lines 37-39 mention the impact of the crucifixion for audiences. The analysis here would benefit from some examples of other representations of the crucifixion in cinema that the author has in mind. Is it correct to say that they ‘fall flat’? There are, for example, figurative representations, which are mentioned later in the draft, that might counter this. One can equally think of the success of the Passion of the Christ (2004) as an example of where the crucifixion impacts a much wider audience (however problematic that representation might be). It might be better to say here that the Jesus film tradition has tended to ornamentalize or sensationalize aspects of the crucifixion but that there are some instances that go beyond this. I can think here of the examples mentioned in the note and of Pasolini’s The Gospel according to Saint Matthew which avoids physicality and emphasizes the reaction of Jesus’ followers and family. This is also important as a film that attempts to transpose (albeit in a different way) the Gospel onto the southern Italian world of the mid-20th century.

Footnote 2: this note references Ocean of Mercy (1978) and it might be useful to include the original title of this film, which is Karunamayudu, and the director Bhim Singh.

Page 2

Line 52 rightly highlights the local setting of the story but while the setting is thoroughly local, the narrative focus is cosmic. The meeting of Jesus and the Satan figure is one of a few key scenes in the film that explicitly invokes the transcendent and signals a meeting of good and evil. Perhaps this point could be developed to include the idea that this is both a local and cosmic scene. Lloyd Baugh’s chapter explores this, and its insights would complement this essay’s approach.

Line 73: A dash here opens a parenthesis (presumably to indicate the scene) but is not closed by a dash but a comma. To give the sentence a greater flow, consider revising its structure. 

Page 3

Line 105: Word missing: not to the extent?

Line 105: Consider rephrasing: . . . where the film truly represents, and is grounded in, an exclusively South African reality.

Line 115: Word choice: perhaps it is better to say 'engaging with them'

Line 117: Word choice: Is 'waffling' an appropriate word here?

Line 118: Instead of ending with a somewhat cryptic 'I think', which occurs on a couple of occasions, perhaps it would be better to spell out why this might be the case. I can think of many reasons from the idea of a 'prophetic fragment,' which might be a helpful idea to develop further, to the notion that the transposition of the Jesus story onto any context is itself an inexact affair given that even the New Testament writings set up a past/present dynamic that is often historically removed from the time of Jesus itself and more rooted in the latter 1st and early 2nd century worlds. Think here of John 8 and the reference to expulsions from the Synagogue - a clearly anachronistic detail taken from the life of the Johannine community. Perhaps something similar or a variation on this inherently NT dynamic is happening here?

Line 121: the author disagrees with Giere's analysis but could explain more how their point 'goes too far'

Page 4

Line 170: The preceding paragraph makes many important points, which I think could be developed further, especially its resonances with the literary gospels. Take for example, some interpretations of Mark, which identify the theme of Rome's totalizing empire and its contrast with Jesus' ministry from the margins. This theme is developed in the book Mark As Story (Rhoads, Michie, and Dewey) and has important resonances with both this film and wider liberation and utopian interpretations of the Christian tradition.

Line 173: consider rewording this sentence to avoid the repetition of the word ‘after’

Line 174: consider rewording to avoid repeating the name Judas

Page 6

Line 238 includes a quotation in bold but I am wondering if the single quotation mark is needed here?

Line 250: should both quotation marks be in bold or normal font?

Page 7

Note 17: this note appears incomplete

Page 8

Line 325: should this be Seneca’s Dial ?

Page 9

Line 377: for conciseness of expression, this should read ‘the horror that slaves and non-citizens felt . . .’

Page 10

Line 412: Why is the soldier’s name, Hundred, in quotation marks? The reference to his role as the centurion, etc. would be better suited to the main text rather than as a parenthesis.

Lines 425-427 mentions the toyi-toyi dance but this would be better embedded into the main text rather than as a bracketed point.

Page 12

In the conclusion, it would be helpful to re-contrast the local and cosmic dimensions. The epilogue not only completes the passion sequence but also resonates with the opening encounter between Jesus and the Satan figure. In that first scene, Jesus states: 'this is my world' - an encapsulation of the incarnation, in a sense - and this scene is a book end of sorts on this prologue. So, it might be useful to note that the universal element of this struggle emerges, inter alia, out of this dynamic.  
